# Surveillance on California dairy farms reveals multiple possible sources of H5N1 influenza virus transmission

A. J. Campbell[1], Meredith Shephard[1☉], Abigail P. Paulos[2☉], Matthew D. Pauly[1], Michelle N. Vu[1], Chloe Stenkamp-Strahm[3], Kaitlyn Bushfield[1], Betsy Hunter-Binns[4], Orlando Sablon[2], Emily E. Bendall[5], William J. Fitzimmons[5], Kayla Brizuela[1], Grace E. Quirk[1], Nirmal Kumar[1], Brian McCluskey[3], Nishit Shetty[6], Linsey C. Marr[7], Jenna J. Guthmiller[8], Jefferson J. S. Santos[9], Scott E. Hensley[9], Edith S. Marshall[10], Kevin Abernathy[4], Adam S. Lauring[5], Blaine T. Melody[11], Marlene K. Wolfe[2], Jason Lombard[3]*, Seema S. Lakdawala �ID [1]*

1 Department of Microbiology and Immunology, Emory University School of Medicine, Atlanta, Georgia, United States of America, 2 Gangarosa Department of Environmental Health, Rollins School of Public Health, Emory University, Atlanta, Georgia, United States of America, 3 Department of Clinical Sciences and AgNext, Colorado State University, Fort Collins, Colorado, United States of America, 4 Milk Producers Council, Ontario, California, United States of America, 5 Division of Infectious Diseases, Department of Internal Medicine, University of Michigan, Ann Arbor, Michigan, United States of America, 6 Department of Civil, Environmental and Architectural Engineering, University of Kansas, Lawrence, Kansas, United States of America, 7 Department of Civil and Environmental Engineering, Virginia Tech, Blacksburg, Virginia, United States of America, 8 Department of Immunology and Microbiology, University of Colorado Anschutz Medical Campus, Aurora, Colorado, United States of America, 9 Department of Microbiology, Perelman School of Medicine, University of Pennsylvania, Philadelphia, Pennsylvania, United States of America, 10 Antimicrobial Use and Stewardship Branch, Animal Health and Food Safety Services Division, California Department of Food and Agriculture, Sacramento, California, United States of America, 11 Lander Veterinary Clinic, Turlock, California, United States of America

☉ MS and APP contributed equally to this work, order was decided by coin flip.
* Jason.Lombard@colostate.edu (JL); seema.s.lakdawala@emory.edu (SSL)

## Abstract

Transmission routes of highly pathogenic H5N1 between cows or to humans remain unclear due to limited data from affected dairy farms. We performed air, farm wastewater, and milk sampling on 14 H5N1-positive dairy farms across two different California regions. Infectious virus was detected in the air in milking parlors and in wastewater streams, while viral RNA was found in exhaled breath of cows. Sequence analysis of infectious H5N1 virus from air and wastewater samples on one farm revealed viral variants relevant for potential human susceptibility. Longitudinal analysis of milk from the individual quarters of cows revealed a high prevalence of subclinical H5N1-positive cows. Additionally, a heterogeneous distribution of infected quarters that maintained a consistent pattern over time was observed, inconsistent with shared milking equipment serving as the sole transmission mode. The presence of subclinically infected cows was further supported by detection of antibodies in the milk of animals that exhibited no clinical signs during the H5N1 outbreak on one farm. Our data highlight additional sources and potential modes of H5N1 transmission on dairy farms.

**Data availability statement:** All data sets generated that support the findings of this study are available on FigShare (https://doi.org/10.6084/m9.figshare.29627840). Raw data from the whole genome sequencing has been deposited to SRA, number PRJNA1416438. The custom Python code used for analysis of droplet size volumes calculated in S1 Fig can be found on FigShare (doi: https://doi.org/10.6084/m9.figshare.29627840).

**Funding:** This work was supported by discretionary funds to S.S.L from Emory University and gift funds to the Emory Center for Transmission of Airborne Pathogens, provided by the California Dairy Research Foundation (https://cdrf.org/) and Flu Lab, a California-based organization founded to advance innovative approaches for the prevention and treatment of influenza (https://theflulab.org/). A.S.L., W.J.F., and E.E.B. were supported by the Michigan Infectious Diseases Genomics Center (NIAID U19 AI181767). This work was also supported in part by a gift to M.K.W. from the Sergey Brin Family Foundation. Part of this publication was made possible, in part, by an Agreement from the United States Department of Agriculture's APHIS to J.L. This publication may not necessarily express the views of APHIS. The funders had no role in study design, data collection and analysis, decision to publish, or preparation of the manuscript.

**Competing interests:** I have read the journal's policy and the authors of this manuscript have the following competing interests: S.S.L. and L.C.M. receive funds from Flu Lab, NIH, NSF, and ARPA-H. A.S.L. receives funds from Flu Lab, NIH, and CDC. A.S.L. receives consulting fees and research support from Roche, outside of the submitted work. S.E.H. is a co-inventor on patents that describe the use of nucleoside-modified mRNA as a vaccine platform. S.E.H reports receiving consulting fees from Sanofi, Pfizer, Lumen, Novavax, and Merck.

**Abbreviations:** BHI, Brain Heart Infusion; BRSV, bovine respiratory syncytial virus; LF, left front; LR, left rear; LVC, Lander Veterinary Clinic; PBS, phosphate-buffered saline; RF, right front; ROIs, Regions of Interest; RR, right rear; RT, room temperature.

## Introduction

Highly pathogenic avian influenza (HPAI) H5N1 clade 2.3.4.4b B3.13 virus was first detected in dairy cattle in March of 2024 and has since spread to 16 states [1–3]. California, the largest dairy-producing state in the US, detected positive herds in August of 2024 [4]. As of September 30, 2025, 771 herds have tested positive in California [5]. High viral loads in milk from H5N1-infected cows, the persistence of H5N1 on milking equipment, and reports of infection following milk-based eye splashes in dairy workers has led to the general consensus that direct contact with unpasteurized milk is a predominant mode of H5N1 transmission between cows and to humans on dairy farms [6–9]. Additionally, environmental sampling conducted on dairy farms with H5N1-infected cows detected the presence of viral RNA on multiple surfaces within the milking parlor, where both cows and humans have closest contact with unpasteurized milk [10,11].

While direct contact with contaminated milk may contribute to H5N1 spread on dairy farms, aerosol-based transmission cannot be ruled out. Aerosolization of unpasteurized milk in dairy milking parlors likely occurs during expression of the first milk, termed forestripping, and other milking procedures. Activities within milking parlors increase the risk of transmission via direct contact with mucus membranes and contamination of surfaces as virus-laden particles settle out of the air [3]. Both naturally infected and experimentally infected cows have been reported to harbor low levels of virus in the nasal cavity [1,7,12–14], which could produce air particles that are capable of spread to other cows or humans. Whether such H5N1 aerosols exist on affected dairy farms and promote transmission is currently unclear, but understanding their role is critical to developing strategies that mitigate all the modes of virus spread on a farm.

Another potential source of H5N1 transmission between cows is the use of reclaimed water contaminated with bucket-collected milk from sick animals and water used to flush milk lines in the parlor after a milking session. Milk from cows with mastitis or other clinical signs does not enter the commercial milk production system and is typically disposed of in an on-farm wastewater system. In this manuscript, any reference to "wastewater" streams refers to the reclaimed farm water collected through these various processes and held in large manure lagoons or sent to fields on farm premises. Reclaimed wastewater can be used at multiple sites on a farm, such as irrigation of fields and flushing of housing pens. Both processes could increase the presence of H5N1 in the environment on and around the dairy and generate virus-laden aerosols. Wastewater in lagoons can also be a water source for migratory birds and peridomestic animals that may become infected through exposure to the virus. Given the many uses of farm wastewater, determining viral loads in these waste streams is vital.

In this study, we describe the results of extensive sampling efforts at 14 H5N1-positive dairy farms. We report detection of virus in the air during milking of suspected H5N1-infected cows on multiple farms and on multiple days, and in the exhaled breath of cows, suggesting that airborne transmission may serve as a mode of H5N1 spread between cows and to humans in dairy parlors. We also detected

H5N1 virus in farm wastewater, which could contribute to the continued spread of H5N1 between cows and to people and other wildlife. Multiple sequence variants were noted from these sources. Finally, we sampled milk from the individual mammary glands (commonly referred to as quarters) of cows on a single farm over several days to determine the distribution of H5N1 positivity over time. We found cows with H5N1-positive milk did not always present with mastitis, a noted clinical sign associated with H5N1 infection [2,15]. In addition, H5N1-specific antibodies in the milk could be detected from cows that never experienced other clinical signs such as drop in milk production. Presentation of mastitis predominated in a specific quarter of the cows while H5N1 quarter positivity varied by animal, but no single quarter was over-represented. The prevalence of infected cows with no clinical signs is indicated both by our data showing H5N1 infection without mastitis and by detection of H5-specific antibodies in biometrically-monitored cows that had no clinical signs during the H5N1 outbreak on a different farm. Together, our data identify multiple potential modes of H5N1 transmission on dairy farms.

## Results

### Detection of H5N1 in the air

During October to December of 2024, five dairy farms in the Central Valley of California identified as having HPAI H5N1 positive cows were enrolled in our initial air sampling studies (S1 Table and Fig 1A for general dairy farm layout). In the first phase of our study, we compared a variety of air sampling strategies during the active outbreak on dairy farms to define the optimal methodology that could be implemented in future sampling efforts. Aggregated milk from cows, referred to as bulk tank milk, was tested frequently for the presence of H5N1 viral RNA from late October 2024 until January 4, 2025 per federal recommendations [16]. The date that each farm tested positive and first observation of clinical signs are depicted in Fig 1B. H5N1 was detected in bulk tanks on farms prior to the observation of clinically ill cows. Ct values from bulk tank on the day of sampling performed in this study are shown in Fig 1C. Three different air sampling devices were tested in this initial phase (Fig 1D and 1E and Materials and methods): (1) open-face PTFE filter cassettes connected to an air sampling pump; (2) Sartorius MD8 Airport; and (3) AirPrep Cub 210. Open-face PTFE filter cassettes were worn on a backpack to model exposure to facility workers. A cone around the MD8 airport or open-face PTFE filter was used to minimize dilution of aerosols and droplets from the source (Fig 1D). On each farm, aerosols and droplets were collected from exhaled breath from individual cows or rows of cows, within milking parlors during milking, and within housing areas using a variety of these aerosol sampling devices. Seventy-one air samples were collected and analyzed for the presence of H5N1 viral RNA (Fig 1E and S2 Table). Six samples tested positive: five with the MD8 Airport sampler and one from an open face PTFE sampler worn during milking in the milking parlor (Fig 1E and S2 Table). Four positive samples were collected in the milking parlor and two, very weakly positive samples, were obtained from the exhaled breath of rows of 15–30 cows in head stanchions (Fig 1F). The portable nature of the MD8 allowed it collect air closer to the milking process (Fig 1D), which may account for the higher detection rate compared to the AirPrep Cub, which was placed stationary within the dairy parlor approximately 14 meters from the milking process. Based on these observations, further studies were conducted primarily using the MD8 Airport sampler on additional dairy farms.

### Infectious aerosols in milking parlors

Based on the successful initial sampling phase, additional surveillance was performed on nine farms, with in-depth focus on three farms in southern California during their outbreaks in February and March 2025 (S1 and S4–S6 Tables and Fig 2A). On these farms, air samples from the milking parlor and exhaled breath of cows in housing pens were collected as previously described, using the MD8 Airport sampler fitted with a cone (Fig 1D). Sampling was conducted within days of each farm testing positive in the California state bulk tank testing program. Bulk tank milk samples from two of the three farms were collected on multiple days by research staff during the indicated time frame, highlighting the viral load of the herd during sampling (Fig 2B; farms EC, EG). The number of cows per herd was recorded for each farm (S1 Table), but the number of infected cows was unknown so sampling was targeted to milking of pens containing cows with known clinical signs.

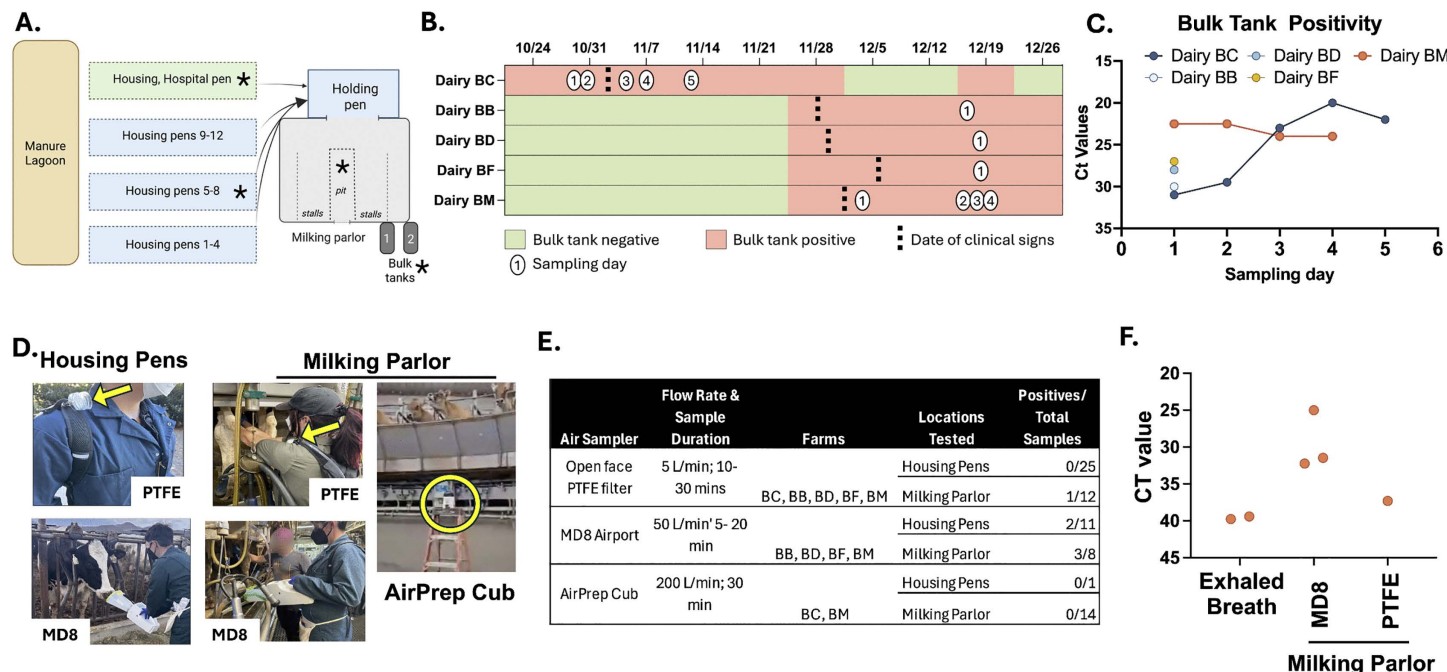

**Fig 1. Successful detection of H5N1 viral RNA in the air in milking parlors in the Central Valley of California using the MD8 Airport sampler.**
**(A)** Top-down schematic of a representative dairy farm layout with multiple pens housed together under open-air structures, a separate hospital pen area, and a holding area where cows are moved prior to milking in the parlor. While the specific layout and structures could vary by farm, the housing pens were typically open-air structures with only a roof, allowing continuous exposure to sunlight and outside air. Holding pens were largely within the indoor dairy parlor structure, with some outdoor air penetrating into the space. The dairy parlor is predominantly an indoor space with air flow occurring through open windows with or without screens. Areas with asterisks denote spaces where air samples were collected. Arrows indicate the movement of cows from housing areas to holding pen, then to the milking parlor. Milk is aggregated into bulk tanks. 'Sick' cows, with clinical signs such as reduced appetite, fever, mastitis, or nasal discharge, were moved to a hospital pen and milked last into individual buckets; their milk was excluded from the bulk tank (for commercial supply) and entered the farm wastewater stream. Schematic created with BioRender.com (*Campbell, A. (2026)* https://BioRender.com/3gmitks). **(B)** Timeline of sampling during initial phase on dairy farms, including date the bulk tank tested positive, bulk tank status, onset of clinical signs in cows, and sampling days numbered 1–5. **(C)** Ct values for H5N1 viral RNA from bulk tank samples on each sampling day. **(D)** Photos depicting each type of air sampler. Left to right, open-face PTFE filters in use on the backpack while in a housing pen and milking parlor as indicated (yellow arrows); the MD8 Airport with a cone, to reduce dilution of the air, sampling the breath of a cow in a housing pen and during milking in the parlor; the AirPrep Cub (yellow circle) was placed either on the ground or was elevated to about six feet above the ground in milking parlors. **(E)** Summary of air sampling methods and H5N1 positive samples from the five dairy farms. **(F)** Ct values (average of duplicates) from the H5N1 positive air samples. Exhaled breath samples also collected using the MD8 Airport sampler.

H5N1 viral RNA was detected in 21 of 35 air samples (Fig 2C; S5 and S6 Tables). Samples are presented as genome copies per liter of air and range from 1 to 10,000. Exhaled breath was collected from rows of cows within the hospital pen on farm EG, and two of 10 samples tested weakly positive with 4–41 genome copies/L of air ($C_{air}$) (Fig 2C). For comparison, other studies of influenza A virus in air around livestock reported concentrations ranging from 7 to 440 gc/L in poultry barns or markets [17–21] and an average of 142–320 gc/L in swine barns [22,23]. Particle sizes generated in milking parlors were quantified using two methodologies, one to capture small aerosols (0.3–25 µm), and larger particles were measured by qualitative analysis of droplets collected on the MD8 gelatin filter (S1 Fig). These results reveal that both submicron aerosols and larger aerosols are likely produced during milking and could contain infectious virus. All positive samples collected on EC and EG were subject to viral titration, and infectious virus was detected in four samples with $C_{air}$ values >1,000 collected with the MD8 Airport in a milking parlor (Fig 2D). Together, these results indicate that H5N1 viral RNA can be found in the air on farms and conclusively demonstrates that infectious H5N1 virus is present in the air during the milking process.

**Fig 2. Presence of infectious H5N1 virus in air samples from dairy farms in southern California. (A)** Timeline of sampling strategy for farms EB, EC, and EG. Light blue dots indicate milking parlor air samples, red dots indicate exhaled breath samples from cows, dark blue dots indicate water samples, and small gray dots indicate sampling dates. Detailed sampling information available in S4–S6 Tables. **(B)** H5N1 positivity in bulk tank milk samples over time from farms EC and EG as determined by viral RNA levels. **(C)** Viral genome concentration in the air ($C_{air}$) calculated as the genome copies (gc) per liter of air (L) from H5N1 positive air samples from the milking parlor (circles) or breath of cows (triangles) for farms EC (green, $N=18$) or EG (purple, $N=4$) over time. No template control samples, processed alongside surveillance samples, were run on every plate and were confirmed to have no amplification. **(D)** Infectious virus from all positive air samples ($N=22$) from farms EC and EG was assessed by plaque assay. Colors are as indicated in **(C)**, open circles denote samples that produced cytopathic effect and H5N1 positivity in TCID$_{50}$ assay but produced no plaques in a separate titration assay. Dotted black line indicates the limit of detection. For B–D, each dot is the average of technical duplicates.

## Detection of infectious H5N1 in farm wastewater sources

We next confirmed that both milking equipment and milk from sick cows on farms EC and EG harbored infectious H5N1 virus during the sampling period (S2 Fig). Since wastewater on farms could potentially be contaminated with H5N1 by these sources, we collected multiple water samples along the reclaimed water stream at each available point such as (1) water moving down the drain in the milking parlor, (2) at the sump pump adjacent to the dairy parlor, (3) fields where captured water is used, and (4) at manure lagoons where reclaimed water is stored (Fig 3A). H5N1 viral RNA was detected at each point of the waste stream, including in manure lagoons that are widely used by migratory birds and in fields with grazing cows (Fig 3B and S3–S6 Tables). Selected samples with an H5N1 RNA concentration exceeding 650 genome copies/mL were subject to viral titration, and two samples contained detectable infectious virus (Fig 3C). These results demonstrate that H5N1 is prevalent in reclaimed farm wastewater sites across dairy farms and may serve as another source of H5N1 spread between cows, to humans, and to peri-domestic animals.

## Presence of H5N1 viral variants in the air

Whole genome sequencing was performed on a subset of air samples and the reclaimed farm wastewater sample with highest genome copies from farm EC (S5 Table), and revealed all samples were within the H5N1 2.3.4.4b B3.13 subclade

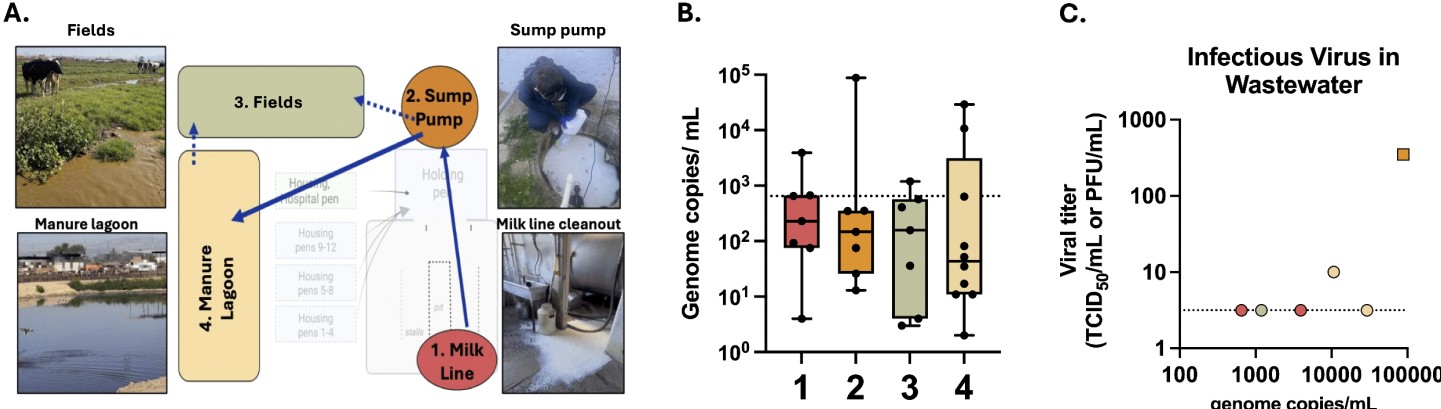

**Fig 3. Detection of H5N1 virus in multiple reclaimed wastewater sites in southern California and Central Valley farms. (A)** Top-down schematic of a representative dairy farm layout with farm wastewater sources depicted and on-site representative pictures included. Sources tested in the schematic match the colors in parts B and C. Schematic created with Biorender.com (*Campbell, A. (2026)* https://BioRender.com/3gmitks). **(B)** Water sources were screened for H5N1 viral RNA by digital droplet PCR and the data is plotted as a box plot. Each dot represents a single sample, solid horizontal lines represent median value, the lower edge is the first quartile, the upper edge is the third quartile, and max and min values are indicated with error bars. Details on location and date of each sample can be found in S3–S6 Tables. Briefly, milk cleanout (1) $n = 7$, sump pump (2) $n = 7$, field (3) $n = 7$, manure lagoon (4) $n = 10$. **(C)** Available samples with genome copies above 650 (indicated by dashed line, $n = 6$) were assessed for infectious H5N1 virus by either TCID$_{50}$ (circles) or plaque assay (square) and infectious titer from either assay is presented.

(Table 1). Variant analysis was conducted for each sample compared to a consensus sequence based on all samples on that farm. To improve the chances of successfully sequencing viral genomes from field samples, nucleic acids extracted from these samples were amplified using three different primer sets. Sequence coverage was variable across the genome and among samples (S3 Fig), and we report only variants that were present in all consensus sequences for a given sample that had coverage at a given locus with at least one consensus sequence derived from a coverage depth greater than 24 reads. Amino acid variants were observed in many samples compared to the farm consensus sequence. Air samples collected from the milking parlor during milking of the same pen of cows on subsequent days revealed variations in the HA, PB2, and PB1 segments (Table 1, compare EC36 to EC47 and EC60). Of note, HA mutations observed in EC36 identified a change at HA position 189 (193 H3 numbering), which is found within the 190-helix, a critical component of the receptor binding site [24,25]. While position 189 does not directly bind to α2,3-linked sialic acids [26,27], mutations at this site are associated with improved binding to human-like α2,6-linked sialic acid [28]. This observation suggests that viral variants with changes in the HA receptor binding site can emerge on farms. Variants were also observed between the air samples collected during milking of sick versus healthy cows on the same day (Table 1, compare EC47 to EC54 and EC60 to EC66). Sequences from an air sample collected in the milking parlor on March 4, 2025 closely matched the sequence observed in the water sample from the same farm on the same day (Tables 1 and S5, samples EC22 and EC27). Taken together, analysis of air and water samples highlight that there is genetic variation on a farm and more detailed analysis of sequences could decipher intra-herd transmission patterns.

## Presence of H5N1 antibodies in subclinical cows

The lack of individual cattle testing prevents detection of subclinical infections. To explore whether H5N1 positivity could be predicted based on a subset of clinical signs, daily milk weight production data was used on Farm FB to bracket cows into three groups: (1) animals that suddenly went dry, likely from H5N1 infection, and were sent to slaughter for beef; (2) those that dropped milk weight and then recovered; and (3) animals without any drop in milk weight (Fig 4A). In April, only cows in groups 2 and 3 were present on farm FB. To assess whether H5N1 infection was observed in subclinical cows,

**Table 1. Consensus variants identified in environmental samples from dairy farm EC.**

| Sample | Date | Source | PB2 | PB1 | PA | HA | NP | NA | M | NS |
|---|---|---|---|---|---|---|---|---|---|---|
| EC22 | 3/4/25 | Air, milking parlor, sick pen cows | | | | | | | | |
| EC27[a] | 3/4/25 | Wastewater, sump pump | | | | | | | | |
| EC36[b] | 3/6/25 | Air, milking parlor, pen 7 cows | A1231G (I411V), A1562C (K521T) | T1533C (syn) | | A613G (N189D), A871G (N275D) | | A834G (syn) | | |
| EC38 | 3/6/25 | Air, milking parlor, pen 8 cows | A1562C (K521T) | | | | | | | G538A (NS1 V180I; NEP syn) |
| EC47[b] | 3/7/25 | Air, milking parlor, pen 7 cows | | C191T (P64L), T1533C (syn) | | | | | | |
| EC54[b] | 3/7/25 | Air, milking parlor, sick pen cows | | | T1449C (syn)[c] | | | A1352G (D451G) | | G672A (NS1 syn; NEP G67E) |
| EC60 | 3/8/25 | Air, milking parlor, pen 7 cows | | | | | | A1352G (D451G) | | G672A (NS1 syn; NEP G67E) |
| EC66 | 3/8/25 | Air, milking parlor, sick pen cows | | T1533C (syn) | | A1271G (N408S) | T1005A (syn) | A233G (syn) | | A427G (NS1 T143A) |
| EC70[b] | 3/9/25 | Air, milking parlor, pen 8 cows | | | T1449C (syn)[c] | | | | | |
| EC71[b] | 3/9/25 | Air, milking parlor, sick pen cows | | | T1449C (syn)[c] | | | | | |

Differences in consensus nucelotide sequences from each sample are indicated relative to the consensus sequence of all environmental samples from farm EC.

Nucleotide positions are relative to the positive-sense gene coding portion of each segment.

Amino acid changes are indicated in parentheses; with numbering relative to the start codon position.

HA amino acid positions are based on the mature HA protein sequence, not the start codon position as suggested in Burke DF and Smith DJ (2014) *PLoS One.*

syn = synonymous mutation.

[a]Only cell-passaged sample was sequenced.

[b]Original and cell-passaged samples were sequenced.

[c]Variant identified only in cell-passaged samples due to no coverage at locus in original samples.

the levels of IgG antibodies targeting H5N1 HA were assessed in expressed milk collected from four recovered cows and 10 animals without any drop in milk weight (Fig 4B). All four recovered animals had detectable H5 antibodies, and six of 10 subclinical cows had detectable H5 antibodies, albeit at a lower level than those observed in the recovered cows. These data suggest that subclinical cows have been exposed to sufficient virus antigen to produce a mucosal immune response within the mammary glands without manifestation of clinical disease.

### H5N1 positivity in individual quarters is heterogenous within a given herd

Contaminated milking equipment is thought to be a primary mode of H5N1 cow-to-cow transmission [9] though definitive proof from natural or experimental infection scenarios is still lacking. There is a specific orientation of the milking equipment when placed on the cow udder such that each teat cup is almost always placed on the same teat or quarter of each cow. Thus, similar quarters might be predicted to be infected throughout a herd given these constraints if equipment was contaminated. To begin to examine the distribution of H5N1 infection in cow quarters, we initiated a longitudinal study on farm EC (Fig 5A). Twenty-three cows were screened for H5N1 positivity; of these, 14 cows were selected for longitudinal sampling, including three cows that screened negative. Milk samples from each individual quarter: left front (LF), right

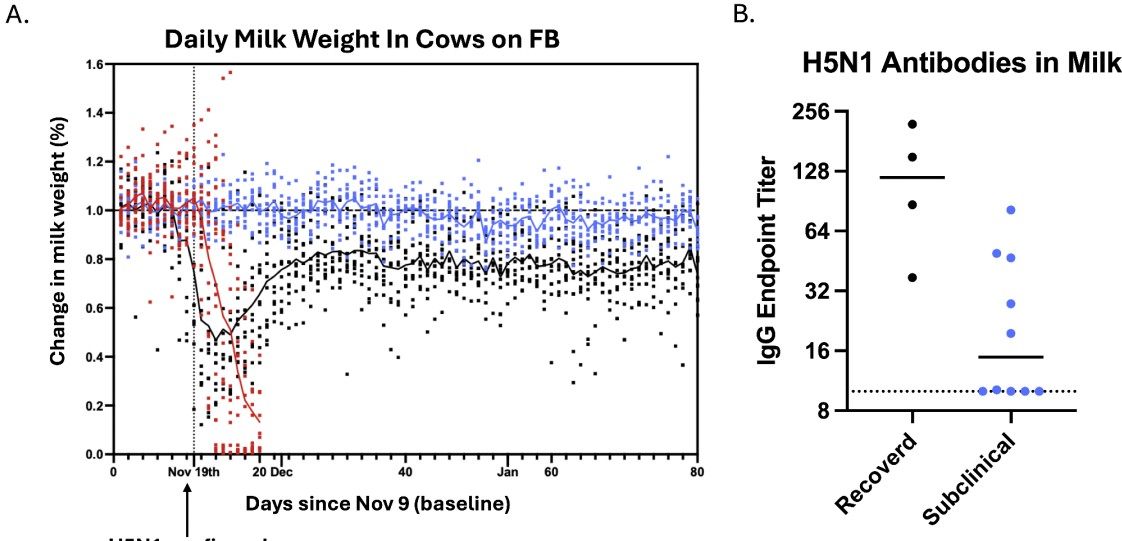

**Fig 4. Presence of anti-H5 HA antibodies in expressed milk from cows with subclinical disease. (A)** A biometric monitoring system on farm FB allowed for assessment of the daily milk weight produced by cows prior to H5 detection in the bulk tank and after. A subset of cows had a significant drop in milk weight that resulted in removal from the herd (red), a subset dropped milk weight and then recovered (black), and finally a subset of animals had less than a 10% milk weight drop (subclinical, blue). Each dot represents data from an individual animal. **(B)** In April 2025, expressed milk was collected from a subset of cows in the recovered and subclinical groups from part A. Anti-H5 HA IgG antibodies were assessed by ELISA and the endpoint titer is shown for 4 cows in the recovered (black group) and 10 cows in the subclinical group (blue).

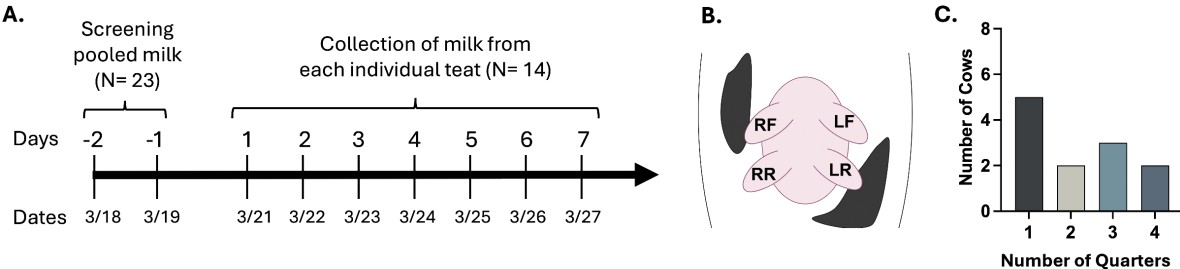

**Fig 5. Longitudinal sampling of individual quarters on cows from farm EC.** Farm EC tested positive on 2/24/2025. **(A)** Timeline of screening and longitudinal sampling (for 7 days, starting 3/20 or 3/21) of individual cows. **(B)** A worm-view schematic identifying the location of each quarter on the udder: right front [RF], right rear [RR], left rear [LR], and left front [LF]. Schematic created with BioRender.com (*Campbell, A. (2026)* https://BioRender.com/3gmitks). **(C)** Number of cows from the longitudinal study with the indicated number of quarters positive for H5N1 viral RNA. Quarters only positive on a single testing day were not included in C, as a single positive could be due to contamination during collection.

front (RF), left rear (LR), and right rear (RR) were collected by a dairy farm worker prior to milking daily over the course of seven days (Fig 5B).

Analysis of individual cows over this timeframe revealed several important features of natural H5N1 infection progression. First, the number of H5N1 positive quarters per animal varied widely, with 5/12 cows having only one quarter positive and the remaining cows having 2–4 quarters positive (Figs 5C and 6). Second, the quarters that tested positive on the first day of sampling generally remained positive throughout the sampling period (Fig 6). This quarter-specific conservation of H5N1 positivity over time is consistent with observations from experimental infections where H5N1 virus remains within the infected quarters and does not spread to uninfected quarters [12,13,15]. Third, there was little variation in the viral

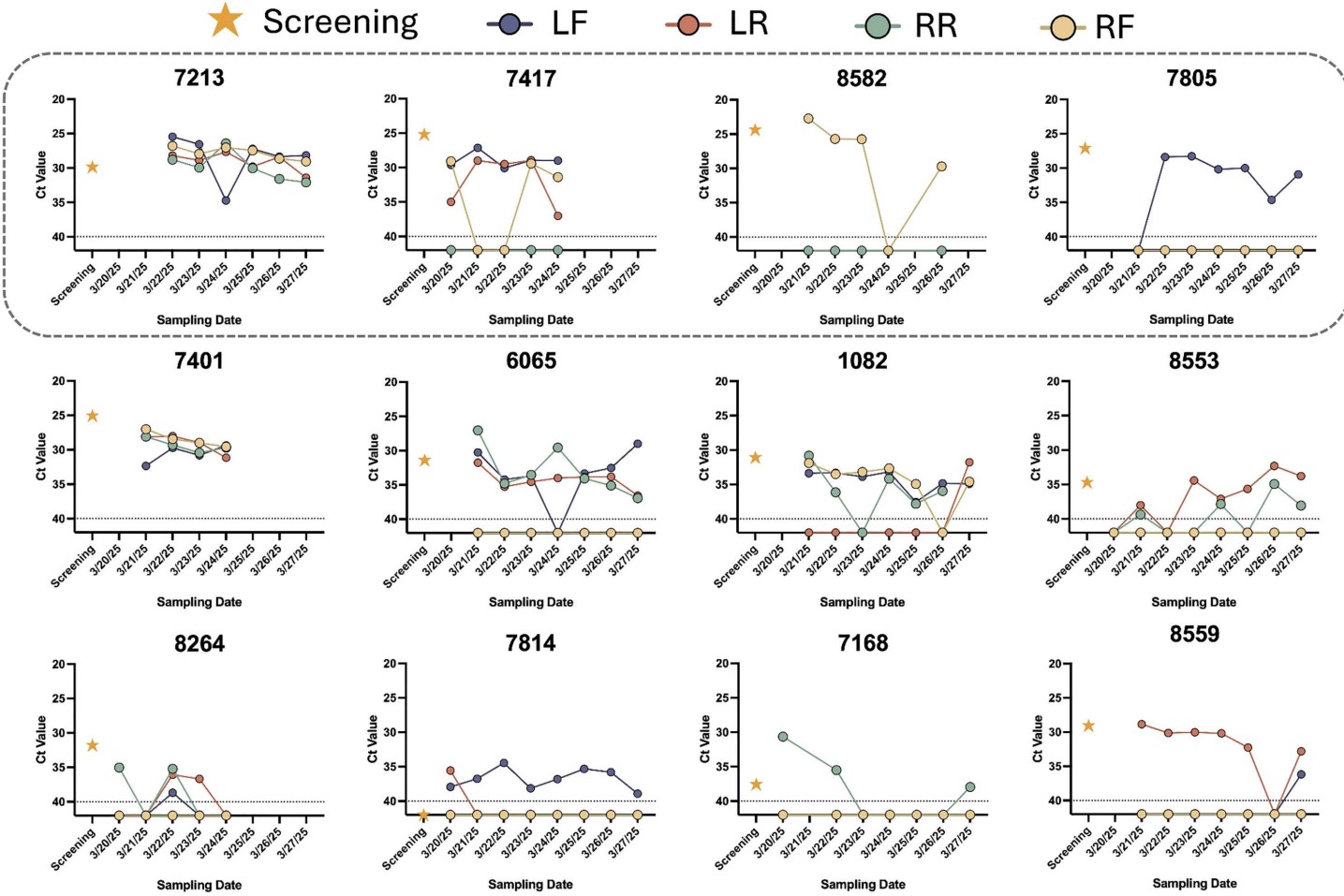

**Fig 6. Heterogeneity in H5N1 positive quarters among naturally infected cows.** H5N1 viral RNA was measured in milk from individual quarters of 14 cows; 12 PCR-positive cows are shown here, including one cow that had originally screened negative (#7814). Ct values averaged from technical duplicates are shown over time for each individual quarter (LF, LR, RR, RF). Yellow stars represent Ct values detected during screening of milk pooled from all 4 quarters of each cow. Cows within the dashed box indicate those that were treated for mastitis during sampling, such that their milk was not put into the bulk tank. Not all cows were available to be tested on all dates. Three cows were sent to slaughter during the sampling efforts: cows #7417, 7401, and 8264.

RNA amount over the 7-day collection period. The absence of H5 viral RNA in specific quarters from infected animals is interesting and may represent quarters that could become infected later during the outbreak, although additional data is needed to confirm this.

H5N1 in dairy cows was identified through investigation of mastitis of unknown origin [2]. Although mastitis (inflammation of the mammary gland that can cause abnormal consistency and discoloration of milk) is often associated with bacterial infections, it has been identified as a common clinical sign in H5N1-infected cows [29]. Since mastitis is usually identified in a single quarter within affected cows [30,31], it is likely that the same quarter would also be positive for H5N1. However, our screening results indicated that mastitis does not always overlap with H5N1 as seven cows were identified as H5N1 positive but lacking clinical signs of mastitis (Fig 7A). We also examined the distribution of mastitis by quarter in a set of cows on farm EC a few weeks after initiation of the H5N1 outbreak but prior to the longitudinal sampling. Within these cows, there was a predominance of mastitis in the RF quarter, with one animal having mastitis in two quarters (RF

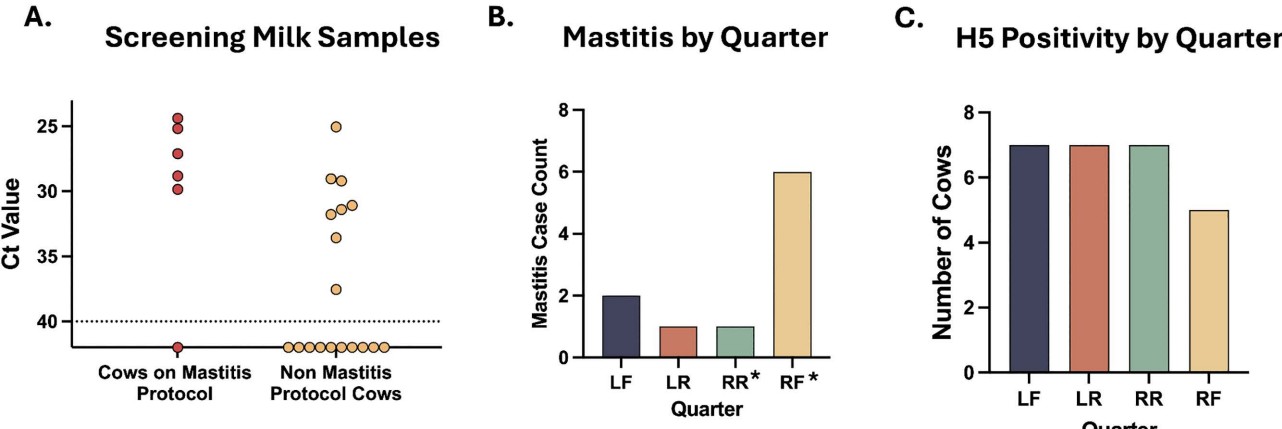

**Fig 7. H5N1 infection does not always correlate with clinical mastitis. (A)** Cows screened for the longitudinal study described in Fig 5, were grouped by whether or not they were treated for mastitis. Each dot represents the Ct value of H5N1 RNA from pooled milk from all four quarters of an individual cow, categorized as either having had mastitis (red, $N=6$) or being randomly selected (yellow, $N=18$), and is the average of technical duplicates. Dotted line represents limit of detection. **(B)** Nine cows on farm EC were noted as having mastitis by milking parlor staff and the affected quarter was recorded by research personnel. These data were collected prior to screening of cows in part A and do not represent the same cows presented in part A or C. The asterisks denote that the cow with mastitis in quarter RR was also positive in RF and was the only cow to present with mastitis in 2 quarters on the farm. **(C)** The 12 cows from Fig 6 were assessed for H5N1 positivity in each quarter. Quarters only positive on a single testing day were not included in C, as a single positive could be due to contamination during collection.

and RR) (Fig 7B). However, this pattern was not observed in the cows from our longitudinal study, where most cows had multiple quarters infected and there was a fairly even distribution of infection among the four quarters (Fig 7C). Although only a small number of cows were sampled, these results suggest that H5N1 positivity can occur in quarters that do not present with mastitis, further supporting the idea suggested by the antibody results (Fig 4) that nonclinical animals may be H5N1 positive.

## Discussion

Elucidating the routes of transmission of H5N1 between cows is critical to defining successful mitigation strategies. In this study, we successfully detected H5N1 in the air and in reclaimed farm wastewater on separate dairy farms on multiple days. This included infectious air samples from three different milking parlors and viral RNA from the exhaled breath of rows of cows on two distinct farms. Additionally, we detected viral RNA in farm wastewater at multiple sites on various farms and infectious virus at two different sites on the same farm. Together, these results highlight the extensive environmental contamination of H5N1 on affected dairy farms and identify additional sources of viral exposure for cows, peridomestic wildlife, and humans. Dairy parlors, which are often enclosed spaces and where aerosolization of milk occurs, pose the greatest threat from inhalation of the virus to dairy farm workers compared to the open-air housing pens.

Surprisingly, after sequencing, we also observed an HA mutation in a residue (H5 HA 189, H3 numbering 193) known to alter sialic acid specificity [28] from an air sample collected during milking of infected cows, suggesting that HA mutations are emerging during this outbreak. Notably, mutations at 189 are associated with improved binding to human α2,6-linked sialic acids [28], and a structure of a 2.3.4.4b H5 with an α2,6-linked sialic acid analog shows that position 189 helps stabilize receptor binding [26]. Whether the N189D mutation observed in this study alters receptor binding and improves tropism to humans remains to be determined. These data support that emerging viral variants may pose a risk to humans working in the dairy parlors through exposure to viruses in the air and on surfaces.

Given the high viral loads of H5N1 found in milk, milking equipment has been implicated as the likely source for viral transmission. However, data from our longitudinal study revealed a heterogeneous pattern of positivity across the subset of cows analyzed, both in the number and location of positive quarters. If milking equipment alone was the driver of cow-to-cow H5N1 transmission it is likely that a similar pattern of positivity would have been observed between the animals. Experimental transmission studies have also failed to induce H5N1 spread between cows using contaminated milking equipment [32]. Thus, alternative mechanisms of H5N1 transmission between cows must exist. The variations in quarter positivity among infected animals may reflect that, and animals with all quarters positive may be those with H5N1 viremia as reported in a subset of cows [33]. Animals with fewer H5 positive quarters may have been infected from contaminated milking equipment (from prior use by infected cows or from aerosols within the parlor) or had damage to their teats that facilitated H5N1 infection via contact with aerosols or contaminated wastewater. Prior reports have suggested that clinical infections are more common in older dairy cows with multiple lactations and later in their lactation cycle [34], which may indicate that there are host factors that could promote susceptibility to H5N1. However, the lack of cattle-level surveillance to define the number of cows becoming H5N1 positive over time on a given farm has limited the ability to define transmission rates of H5N1 between cows.

Surprisingly, we observed H5N1 positivity was not always associated with mastitis, suggesting many subclinical infected cows may be present on a farm. Additional evidence supporting subclinical H5N1 infections on dairy farms was obtained through assessment of H5-specific antibodies in animals without a drop in milk weight production on a different farm (FB) with an outbreak that occurred over 4 months earlier. Assessment of mastitis-affected quarters in a distinct set of cows from the same farm done prior to initiation of the longitudinal study identified a clear preference for the RF quarter. In contrast, analysis of the five cows with only a single H5N1 positive quarter revealed two animals with a positive LF quarter, one LR, one RR, and one RF. The lack of a consistent quarter infected across all cows, as observed for mastitis, suggests that these two features are not linked and may have distinct modes of transmission. The over-representation of RF in mastitis presentation in this study could also be due to performance and efficiency of the milking equipment or other mechanical issues specific to this farm, none of which were assessed during this study.

While field surveillance is critical to understanding ongoing infectious disease outbreaks, there are many limitations that are important to note when considering the data presented in this manuscript. First, longitudinal sampling was conducted on a single farm with 14 cows over a short period of time, which may have missed patterns that could be revealed with a larger number of animals. Additionally, sampling across multiple farms, for longer periods of time, or earlier in an outbreak may reveal different patterns. Due to constraints on time and personnel, shared milking equipment could not be traced in our study, so patterns of milking equipment usage and order of milking for all 14 longitudinal animals are not known and likely varied each day of sampling. Further examination of antibody load and deep sequencing of virus from individual quarter samples could reveal the phase of the viral infection in the animal and provide a glimpse into transmission chains. Second, although previous studies have shown H5N1 influenza virus to remain infectious for days to weeks in aquatic environments [35], the limited number of samples with infectious virus detected in our reclaimed farm wastewater samples could be a consequence of sample fragility, sample complexity, and/or downstream processing inefficiencies. Thus, infectivity in the wild could vary and may be underestimated in our study. However, since high infectious titers of H5N1 are present in milk [7] and milk may enhance the stability of H5N1 as we reported previously [9], this could explain why we were able to capture some infectious virus in the waste stream sampling. Similarly, the amount of infectious virus we detected in air is likely an underestimation based on the collection efficiency and sampling processing issues that are known to make air sampling challenging in real-life settings. Lastly, all air samples from breath of individual cows were H5N1 negative, including from individual cows identified by a veterinarian or other trained professional as demonstrating obvious clinical signs suggestive of H5N1 infection. However, in these instances H5N1 infection was not always confirmed at the time of sampling since rapid detection of H5N1-positive cows on a farm is not yet feasible. Remedies for these limitations would include more longitudinal sampling of individual cows on dairy farms during outbreaks, including

assessment of individual quarters for H5N1 and mastitis, daily nasal swabs and respiratory emissions, as well as tracing of milking equipment. Individual cow sampling would allow for an assessment of the transmission efficiency and secondary attack rate of H5N1 in cows in a herd and a correlation between viral load and clinical onset. Additionally, a rapid diagnostic system that can identify H5N1-positive cows during milking is critical to isolating infected animals and facilitating field sampling [3].

Taken together, our data confirm the presence of infectious H5N1 virus in the air and reclaimed farm wastewater sites. In addition, we observed high viral loads and H5 antibodies in the milk of cows, including those without clinical signs, and heterogenous patterns of H5N1 positivity by quarter, suggesting that multiple modes of H5N1 transmission likely exist on farms. These transmission routes could include contaminated milking equipment from an infected cow, aerosols generated within the milking parlor, and/or contact of teats with contaminated water used to clean housing pens. Multiple mitigation strategies should therefore be implemented to reduce the risk of H5N1 spread within a herd and to humans. Respiratory and ocular personal protective equipment (PPE) for farm workers to prevent deposition of virus-laden aerosols on these sites, especially in the milking parlor. Disinfection of milking equipment between milking of each cow, such as with consistent use of backflush system, could also reduce spread of H5N1 between cows. Treatment of milk from sick cows to inactivate H5N1 prior to disposal as well as treatment of waste streams prior to their use in fields or on farms should also be considered. Finally, identification of infected cows, regardless of clinical signs, for isolation will help reduce the transmission of H5N1 on farms.

## Materials and methods

### Ethics statement

Work performed by Emory researchers was deemed to be exempt by the Emory University Institutional Animal Care and Use Committee. Work performed by Dr. Lombard's team was deemed to be exempt by the Colorado State University Institutional Animal Care and Use Committee. All milk samples, including individual quarter and composite samples, were collected by farm workers during normal farm procedures and passed to research staff. Analysis of infectious viral titer was conducted in a BSL3 laboratory at Emory University under protocols approved by the Emory Institutional Biosafety Committee.

### Dairy farm details

Initial sampling was conducted on five dairy farms in the Central Valley of California from October through December 2024; then subsequently on seven farms in southern California, and on two additional farms in the Central Valley from February to April 2025. Each of these dairy farms had confirmed H5N1 infections in cows as determined by screening of on-dairy bulk tank milk samples in accordance with the National Milk Testing Strategy Federal Order from December 2024. Bulk tank samples taken by our group were collected using either a clean metal dipper, a spigot in the tank, or aseptically with the needle-only QualiTru system into 50mL conical tubes, depending upon the setup of each dairy. The sizes and infrastructure of surveyed dairies varied and are detailed in S1 Table. Determinations of clinical signs and mastitis were made by dairy farm owners, workers, and/or veterinarians, depending on each dairy.

### Air sampling devices and processing

Air samples were collected at sources including exhaled breath of individual cows and rows of cows in housing pens (15–30 cows per filter), within milking parlors during milking, and sites where wastewater is exposed to the environment such as open sump pumps, manure lagoons, and fields. The air samplers used in this study are described below. Details of sampling for each source on each farm can be found in S2–S6 Tables.

**PTFE open-face filter and GilAir 5 pump.** The GilAir 5 Personal Air Sampler pump (Gilian) was set to an air flow rate of 5L/min, the highest rate for this device, during all sampling periods. Aerosols were collected onto Fluoropore

37 mm diameter, 3 µm pore size PTFE membrane filters (Millipore Sigma Ref. FSLW03700) housed within open-faced plastic cassettes. Such filters have been shown to have a physical collection efficiency of ~95% for MS2, a bacteriophage that is smaller than influenza virus [36]. The GilAir 5 pump was worn in a backpack, with tubing connecting the pump to the filter cassette, which was either fastened onto the shoulder strap facing outward or was housed within a plastic cone to enable targeted sampling of cows or processes (Fig 1D). PTFE filters from farm BC were put into 50 mL conical tubes with 10–15 mL Brain Heart Infusion (BHI) at the end of each sampling day, vortexed for 30 s, and shipped at 4 °C overnight to the Iowa State Veterinary Diagnostic Laboratory, where they were assayed for H5N1 viral RNA per standard operating procedures and NAHLN protocols. PTFE filters from farms BB, BD, BF, and BM were processed at the end of each sampling day at the Lander Veterinary Clinic (LVC). Filters were placed into 50 mL conical tubes with 750 µl of VTM (BD Ref. 220526) and vortexed for 60 s, ensuring the entirety of the filter was coated by VTM. Tubes were centrifuged for 10 min at 1,000 rpm, and samples were stored at 4 °C until RNA extraction and RT-qPCR at LVC as described below. PTFE filters from 2/26/25 to 4/3/25 were placed into 50 mL conical tubes with 1 mL of VTM (BD Ref. 220526), ensuring VTM coated the filter, and then stored at 4 °C until shipment and processing at Emory University. The conical tubes containing the filters were centrifuged for 10 min at 1,000 rpm and media removed for RNA extraction; the remaining sample was stored at −20 °C.

**MD8 airport sampler.** The MD8 Airport (Sartorius) is a handheld air sampler that was set to an air flow rate of 50 L/min for all sampling periods. The sampler used 80 mm diameter gelatin filters, which are designed to facilitate capture of pathogens while maintaining viability. The MD8 was always used with a plastic cone fitted onto the front to minimize dilution of aerosols from the source and to protect the gelatin cassette from direct contact with larger liquid droplets. When used around wastewater, the MD8 was held as close as possible to a source of aerosolization, where water was most agitated or the point of impact of source wastewater into its sink destination. To reach such points that could not be approached closely enough on foot, the MD8 was attached to the end of a 25-foot extendable metal pole for sampling. MD8 gelatin filters from BB, BD, BF, and BM were processed at the LVC at the end of each sampling day. Filters from all other farms were shipped to and processed at Emory University. All filters were processed using the same protocol. Briefly, filters were carefully broken up using sterile forceps and placed into 50 mL conical tubes with either 3 mL of VTM or 1.5 mL VTM and 1.5 mL phosphate-buffered saline (PBS). Tubes were vortexed for 60 s to ensure all pieces of the filter were in contact with the liquid media. To fully dissolve the gelatin filter, tubes were placed in a warm water bath for 10 min. After, an aliquot was removed for RNA extraction while the remaining sample was stored at 4 °C (LVC) or −20 °C (Emory).

**AirPrep Cub 210 air sampler.** The AirPrep Cub (InnovaPrep), with its higher air flow rate of 200 L/min, was used to sample ambient air in environments where groups of sick and potentially H5N1-positive cows gathered, such as the milking parlor and housing pens. The sampler was placed either on the ground or at a height of ~2 m and set to sample for 60-min intervals. Used filters from the AirPrep Cub were put into 50 mL conical tubes with 10–15 mL of BHI media (supplied by National Veterinary Services Laboratories, NVSL) at the end of each sampling day. Tubes were vortexed for 30 s and shipped overnight at 4 °C to the National Animal Health Laboratory Network (NAHLN) lab at Iowa State University, where they were assayed for H5N1 viral RNA per standard operating procedures and NAHLN protocols.

**Estimation of particle sizes and numbers in milking parlor air.** An AeroTrak Handheld Airborne Particle Counter Model 9306 (TSI) was used to determine the sizes and number of particles in the milking parlor of dairy farm EC. Readings were collected for ~10 min while following the parlor worker in the pit during the milking process. Each interval corresponded to the total time it took to milk the cows on one side of the parlor, from the time the cows entered until they were released from the milking stalls. The AeroTrak reported particle counts in six size bins: 0.3–0.5 µm, 0.5–1 µm, 1–3 µm, 3–5 µm, 5–10 µm, and 10–25 µm while sampling at a flow rate of 2.83 L/min. Sampling with the AeroTrak was done on two consecutive days on dairy farm EC.

## Surface swab collection and processing

The interior surfaces of milking liners as well as their exterior shells were sampled with pre-assembled swab kits (Becton Dickinson Ref. 220526). For milking unit shells, the flocked swab was unwrapped, pre-moistened by briefly dipping into the included tube of viral transport media (VTM), pressed and dragged over the surface, sealed into the tubes, and bagged for subsequent processing. Pre-assembled swab tubes (BD Ref. 220526) containing surface samples were processed using the same protocol at LVC and Emory University. Tubes were vortexed for 10 s with the flocked swab inside, then centrifuged for roughly 10 s at 1,000 rpm to collect all liquid at the bottom of the tube. An aliquot was then removed for RNA extraction, and the remaining sample was stored at 4 °C (LVC) or −20 °C (Emory).

## Wastewater sample collection and processing

Wastewater samples were collected from multiple locations on each farm (see S3–S6 Tables). Whenever possible, wastewater from the beginning of the milk line cleanout was collected at the end of each milking shift. Samples were also collected from sump pumps, manure lagoons, and/or fields whenever possible. Wastewater samples were collected aseptically. When waste flows were within reach, the external sides of 50 mL falcon tubes or 250 mL and 1 L bottles were first sterilized with 10% bleach and then lowered into the water stream until tubes were filled. When waste flows were out of reach, a 25 ft pole with a bottle adapter (VWR) and 1 L bottles were used for sample collection. The base of the pole, bottle adapter, and a zip tie were sterilized by submerging them in 10% bleach for 5 min before sample collection. The outside of the bottle was also wiped with 10% bleach to sterilize and was then attached to the adapter with the zip tie. The sample was collected by dipping the bottle into the wastewater until full; full bottles were wiped with 10% bleach again and maintained at 4 °C from time of collection until processing at Emory University.

## Quantifying H5N1 viral RNA

**RNA extraction and RT-qPCR.** Samples presented in Fig 1 were processed at LVC. For samples processed at LVC, RNA extraction was carried out using 200–400 µl of sample, using the MagMax CORE Nucleic Acid Purification Kit (Applied Biosystems, Cat. A32700) on the KingFisher Flex 96 system (ThermoFisher) followed by assaying with the Swine Influenza Virus RNA Test Kit (Applied Biosystems, Cat. 4415200). Eight microliters of extracted RNA samples were used in RT-qPCR reactions with the QuantStudio 5 Real-Time PCR System (ThermoFisher, Cat. A28569) according to manufacturer's instructions, with the modification of increased cycling from 40 to 45 to better visualize late-cycle amplification. A no-template control was added to each plate to ensure that any samples with high CT values were not due to background issues. Samples from all other figures were processed and analyzed at Emory University. Prior to processing, all samples received at Emory BSL3 lab were aliquoted for RNA extraction (160 µL for RNA extraction) so that they were not handled while other studies were being processed. Lysis buffer was then added to the pre-aliquoted samples in BSL3 and subsequent nucleic acid extraction was performed in the BSL2 lab (per approved biosafety protocols) using the NucleoMag RNA kit (Machery-Nagel) on an EpMotion 5075 (Eppendorf), with inclusion of a negative extraction control (PBS) in each extraction run. Viral detection required both sample replicates to each exhibit amplification (have CT values) using H5 HA-specific primers TATAGARGGAGGATGGCAGG and ACDGCCTCAAAYTGAGTGTT and probe FAM-AGGGGAGTGGKTACGCTGCRGAC- Black hole quencher, with the iTaq Universal Probes One-Step kit (Bio-Rad, Cat. No. 1725141) on a CFX Connect real-time PCR system (Bio-Rad) using published primers and cycling conditions [37,38]. A no-template PCR control (water) was also included on each plate. No amplification (Ct values) was ever observed for any negative extraction control (PBS) or no template PCR control (water) wells on any of the RNA extraction or RT-qPCR plates processed (greater than 10 in total). All plates included serial dilutions of an in vitro transcribed H5 HA RNA standard with a known concentration for quantification of genome copies. Samples with a CT value >38 were reextracted and run again to confirm positivity. A subset of available samples tested at LVC were sent to Emory for comparison of the qPCR methods and similar results were obtained.

**Wastewater sample processing and ddPCR.** Bottles with wastewater samples were wiped down with 70% ethanol prior to opening, and then approximately 45 mL of raw sample was centrifuged at 16,500$g$ for 5 min to remove large solids and debris. Ten milliliters of the supernatant was input into a concentration step using NanoTrap Microbiome Particles A and B (Ceres Nanosciences) in a KingFisher Apex (ThermoFisher); an endogenous control bovine respiratory syncytial virus (BRSV) was added to each sample prior to concentration. Total nucleic acids were extracted using the standard KingFisher MagMax Viral Pathogen kit (ThermoFisher) and eluted in 100 µl. The ddPCR assays targeting the influenza A M gene (total influenza A) and hemagglutinin H5 were run in duplex on all wastewater samples according to previously published protocols [39]. Briefly, droplets were generated using an AutoDG (Bio-Rad), targets amplified using a thermal cycler (Bio-Rad), and quantified using the QX600 (Bio-Rad). Three ddPCR replicates were performed per sample, and the three wells were merged for analysis. A sample was considered positive for a target when three or more droplets were positive. To ensure extraction efficiency, BRSV was quantified in all samples using a singleplex ddPCR assay on a single well [3].

## Infectious virus titration

Samples were tested for infectious virus within Emory BSL3 laboratory with approved protocols. Samples were titrated on confluent (MDCK) epithelial cells (ATCC) by plaque assay as previously described [40] or tissue culture infectious dose 50 (TCID$_{50}$) assay using the Reed and Muench method [41]. Any air samples positive for H5N1 viral RNA determined by RT-qPCR were assayed for presence of infectious virus as described above. All positive milk samples from cows in the longitudinal study were tested for infectivity, as well as positive milk samples collected from all four quarters of individual cows. All four positive surface swabs of milking unit shells as well as positive swabs of milking inflation liners were assayed for infectivity. Six positive wastewater samples were selected for infectivity testing with both TCID$_{50}$ and plaque assay based on having genome copy per milliliter concentrations above 650 and location of sample collection. All titration studies were done with infection media (1× Minimal Essential Media, Corning Ref. 15-010-CV; with 4mM L-Glutamine, Sigma Aldrich Ref. G7513; and 1:1,000 TPCK Trypsin, Worthington Biochemical Ref. LS003750) containing double the concentration of antibiotics and antimycotics (4× final concentration of Antibiotic-Antimycotic 100×, Gibco ref. 15240062) to reduce bacterial and fungal contaminants in the environmental samples. In many cases both assay types were used to determine infectious virus titration, since TCID$_{50}$ method allows for complementation of semi-infectious virus particles. The highest virus titer for a given sample is reported. In cases with ambiguous cytopathic effect, additional assays were performed to verify positivity, including agglutination of chicken red blood cells with cell supernatant and/or serial passage of 20 µL of the cell supernatant onto fresh MDCK cells to assess for presence of virus as compared to other environmental contaminants. In addition, supernatants from air samples that were positive were RNA extracted for sequencing efforts to identify viral variants.

## Calculation of virus concentration per liter of air

All RT-qPCR plates processed at Emory University contained a dilution series of an in vitro transcribed RNA product of known copy number, allowing generation of a standard curve. This was used for absolute quantification of the genome copy numbers present in each reaction and per milliliter. The genome copy numbers per milliliter were then used to estimate the number of genome copies present per liter of air; the volume of air sampled, in liters: $V_{air}$ = *air flow rate (LPM)* × *sample duration (minutes).* The virus concentration per liter of air was calculated as:

$$C_{air} = \frac{RT-qPCR \ concentration \ \left(\frac{gc}{mL}\right) \ \times \ RNA \ elution \ volume \ (mL) \ \times \ \left(\frac{Total \ volume \ used \ to \ process \ sample \ (mL)}{Volume \ of \ sample \ used \ in \ RNA \ extraction \ (mL)}\right)}{V_{air}}$$

## Estimation of aerosolized droplet sizes deposited on MD8 Airport filters

Photos were taken of 14 MD8 Airport gelatin filters immediately after air sampling. These photos were analyzed to determine the size of aerosolized droplets deposited on the filters using ImageJ. Photos were cropped to include only the filter surface area, then rescaled to 800×800 pixels to standardize and allow size conversion from pixels to millimeters since MD8 gelatin filters have 80 mm diameter. To detect spots, a Gaussian blur ($\sigma = 2$ pixels) was applied to reduce noise with the "Find Maxima" function to generate a mask from the detected maxima. The Regions of Interest (ROIs) were selected with a circularity filter applied (>0.5) to select only spherical spots and the ROIs were restored onto the original unmasked image for accurate pixel measurement. All detected ROIs were manually inspected to remove false positives and artifacts from maxima detection. The ROI measurement function was used to calculate the number of spots and their area (pixel$^2$), and the data was exported as a.csv file for processing through a custom Python script (available at FigShare, https://doi.org/10.6084/m9.figshare.29627840) to convert spots from pixels$^2$ to mm$^2$ and calculate spot volumes.

## Whole genome sequencing analysis

Whole genome sequencing was performed on nine air samples and one wastewater sample collected from farm EC (Table 1). For five of the air samples, a cell-passaged viral population was sequenced in addition to the original sample. For the wastewater sample, only a cell-passaged viral population was sequenced. Nucleic acids were extracted using the NucleoMag RNA kit (Machery-Nagel) with an epMotion 5075 (Eppendorf). The 8 genomic segments of IAV were amplified in a one-step multiplex RT-PCR [42]. For each specimen, extracted RNA, was amplified with each of three different primer sets: Uni12/Inf1, Uni12/Inf3, Uni13/Inf1 [42]; MBTuni-12, MBTuni-12.4, MBTuni-13 [43]; OptiF1, OptiF2, OptiR [44]. cDNA samples were prepared for sequencing using the Oxford Nanopore Technologies (ONT) platform. Unique barcodes were ligated to RT-PCR products from each sample using end repair and ligation kits (New England Biolabs) and then pooled. These pools were ligated to pore adaptors and sequenced with a GridION instrument. Sequence reads were aligned to the nearest IAV reference and sample consensus sequences were generated using IRMA [45]. Influenza virus clades were assigned based on hemagglutinin sequences using Nextclade [46]. The sample consensus sequences from each primer set were aligned using MUSCLE in MegAlign Pro (DNA Star). The overall consensus sequence from all samples sequenced in this study was used as a reference to identify variants in the individual sample consensus sequences from each primer set. Only mutations found in all of a sample's consensus sequences that had read coverage at that genetic locus are reported. All reported mutations were identified at loci with 24-fold or greater coverage depth from at least one of the primer sets.

## Longitudinal study of infected cows

Longitudinal sampling was conducted on dairy farm EC from 3/18/25 to 3/27/25. Cows were screened for H5N1 virus infection on 3/18–3/19 based on being in treatment for mastitis (as determined by dairy farm parlor workers) or were randomly selected during milking. Milk from cows on mastitis treatment doesn't go into the bulk tank. A composite sample of milk was collected from all four quarters and shipped overnight at 4 °C to Emory University for RNA extraction and RT-qPCR as described above. Screening results informed the selection of cows for inclusion in the study group; selected cows were marked on right and left flanks to enable identification in pens and the milking parlor, and identifications were confirmed with cows' ear tag numbers. Eleven H5-positive cows and three H5-negative cows were selected for the study group. One cow (7,814) was selected with a negative sample at screening, but on every subsequent study day tested positive for H5N1. Daily, from 3/20/25 to 3/27/25, a sample of milk from each quarter of the selected cows was collected into separate 50 mL conical tubes and stored at 4 °C until shipping and processing at Emory University for viral RNA extraction, RT-qPCR, and/or virus titering as detailed above. Some cows from this study were not available on all testing days. Sampling of the exhaled breath of all available cows on study on a single MD8 Airport gelatin filter was done

on 3/19, 3/22, and 3/24. All air filter samples were stored at 4 °C until they could be processed at Emory University and stored at −20 °C prior to RNA extraction and RT-qPCR as described above.

**Antibody detection in cows post-H5N1 outbreak**

**Data collection and analysis.** Cows on farm FB are fitted with ankle monitors (Afimilk) that record biometric data and pair with milking units during operation to record data on each animal's expressed milk. In November 2024, farm management used this information to assess clinical signs and assign "sick" status to cows suspected to be infected with H5N1. Biometric data records were obtained for 40 total animals assigned into 3 groups by farm management: (1) those culled from the herd due to signs of suspected H5 infection, (2) those that had shown signs of H5 infection but recovered, and (3) those that never showed signs of infection. Since milk weight was the primary clinical sign used by the farm to identify suspected infections, this information was used to examine differences in the 3 groups of cows above during and after the farm's outbreak.

**Enzyme-linked immunosorbent assay (ELISA).** Milk from 4 cows in the "Recovered" group and 10 cows in the "No Signs" group from farm FB was collected by farm management in April 2025 and shipped overnight on ice to Emory. Frozen aliquots were shipped on dry ice to the University of Pennsylvania for ELISA. ELISA was performed on 96-well Immulon 4HBX extra high binding flat-bottom plates (Thermo Fisher, Waltham, MA). Plates were coated with PBS or recombinant HA protein from A/dairy cattle/Texas/24-008749-002-v/2024(H5N1) at 2 μg/mL in DPBS at 4 °C overnight. Plates were washed three times with 1× PBS + 0.1% Tween-20 and then blocked for 1 h at room temperature (RT) using blocking buffer (1× TBS, 0.05% Tween-20, and 1% bovine serum albumin). Bovine milk samples were serially diluted 2-fold in dilution buffer (1× TBS, 0.05% Tween-20, and 0.1% bovine serum albumin). After blocking, plates were washed three times with 1× PBS + 0.1% Tween-20. Diluted milk samples were then transferred to ELISA plates and incubated for 2 h at RT. After washing three times with 1× PBS + 0.1% Tween-20, peroxidase-conjugated mouse anti-bovine IgG monoclonal antibody (clone IL-A2; Bio-Rad) was added to corresponding wells at a concentration of 1:1,000 in dilution buffer for 1 hr at RT. After washing three times with 1× PBS + 0.1% Tween-20, all plates were developed by adding SureBlue TMB peroxidase substrate (SeraCare, Gaithersburg, MD) for 5 min at RT, followed by stopping the reaction with 250 mM HCl solution. The absorbance was measured at 450 nm using a SpectraMax ABS Plus plate reader (Molecular Devices, San Jose, CA). Background OD values from the plates coated with PBS were subtracted from the OD values from plates coated with recombinant HA protein. A dilution series of the human IgG monoclonal antibody Fi6v3, which is reactive to the A/dairy cattle/Texas/24-008749-002-v/2024(H5N1) recombinant HA protein, was included on each plate as a control to adjust for interassay variability. An anti-human IgG-HRP secondary antibody (Jackson ImmunoResearch Laboratories) was added to these standardization wells. Fi6v3 monoclonal antibody was used to set the OD threshold on each plate and to ensure that the same OD threshold was used on all plates.

## Supporting information

**S1 Fig. Quantification of aerosol and droplet sizes captured in milking parlors. (A)** A representative MD8 gelatin filter with aerosol and droplet marks. Images from milking parlor sampling were collected on multiple days and processed to quantify the aerosol sizes. Processing included cropping of images, blurring, and thresholding of pixel intensities to identify individual spots. **(B)** Analysis of 14 distinct MD8 filters provided 843 unique spots that were analyzed for the volume of each droplet using an equation to estimate the spread factor (see Materials and methods). **(C)** Quantitation of number of aerosol particle counts of different sizes within the milking parlor at farm EC on the indicated sampling dates. Four different measurements were collected on 3/28.
(TIF)

**S2 Fig. Viral loads from milking equipment surface samples and from sick cow milk samples collected prior to initiation of the longitudinal study (prior to 3/18/25).** Infectious virus titers as determined by plaque assay compared to viral RNA levels (Ct values) for **(A)** swab samples collected from milking equipment or **(B)** milk collected from sick cows, whose milk would go into buckets and not the bulk tank, on farms EC (green dots) and EG (purple dots).
(TIF)

**S3 Fig. Coverage plots for sequencing of environmental samples from dairy farms.** The sequencing read coverage across the concatenated influenza A virus genome for each sample listed in Table 1 is presented in its own plot. Samples sequenced from cell-passaged virus are denoted CP1 for "cell passage 1". The concatenated genome includes PB2 (positions 1–2,314), PB1 (positions 2,315–4,631), PA (positions 4,632–6,840), HA (positions 6,841–8,587), NP (positions 8,588–10,094), NA (positions 10,095–11,525), M (positions 11,526–12,527), and NS (positions 12,528–13,388). Positions with coverage greater than 250 reads were assigned values of 250 on these plots for ease of visualization. Three different primer sets (Uni: Uni12/Inf1, Uni12/Inf3, and Uni13/Inf1 [42]; MBT: MBTuni-12, MBTuni-12.4, and MBTuni-13 [43]; and Opti: OptiF1, OptiF2, and OptiR [44]) were used to amplify viral genomes prior to sequencing. Variants were only reported if present in the consensus sequence from all primer sets that had coverage at that locus with at least one primer set yielding a coverage depth of at least 24 reads.
(TIF)

**S1 Table. Details of 14 dairy farms that were sampled in California.**
(PDF)

**S2 Table. Details of air sampling during the initial phase on farms in the Central Valley of California in late 2024.**
(PDF)

**S3 Table. Air sampling details for dairies with 1 or less positive environmental samples, Feb–Apr 2025.**
(PDF)

**S4 Table. Sampling details for dairy farm EB during spring 2025.**
(PDF)

**S5 Table. Sampling details for dairy farm EC during spring 2025.**
(PDF)

**S6 Table. Sampling details for dairy farm EG during spring 2025.**
(PDF)

## Acknowledgments

We thank the farmers and members of the dairy industry who allowed us to conduct these studies on their farms and supported our efforts. Thanks to the team at California Department of Food and Agriculture for research support and coordination. We also thank Dr. Montserrat Torremorell for use of the AirPrep Cub 210, Dr. Rachel Duron for editorial assistance, and members of the Lakdawala lab for critical review.

## Author contributions

**Conceptualization:** A. J. Campbell, Matthew D. Pauly, Marlene K. Wolfe, Jason Lombard, Seema S. Lakdawala.

**Data curation:** A. J. Campbell, Meredith Shephard, Abigail P. Paulos, Chloe Stenkamp-Strahm, Kaitlyn Bushfield, Emily E. Bendall, William J. Fitzimmons, Grace E. Quirk.

**Formal analysis:** A. J. Campbell, Meredith Shephard, Abigail P. Paulos, Matthew D. Pauly, Michelle N. Vu, Chloe Stenkamp-Strahm, Orlando Sablon, Emily E. Bendall, William J. Fitzimmons, Kayla Brizuela, Grace E. Quirk, Nirmal Kumar, Brian McCluskey, Jefferson J. S. Santos, Blaine T. Melody, Seema S. Lakdawala.

**Funding acquisition:** Adam S. Lauring, Marlene K. Wolfe, Jason Lombard, Seema S. Lakdawala.

**Investigation:** A. J. Campbell, Meredith Shephard, Abigail P. Paulos, Chloe Stenkamp-Strahm, Jenna J. Guthmiller, Jefferson J. S. Santos, Blaine T. Melody, Jason Lombard, Seema S. Lakdawala.

**Methodology:** Michelle N. Vu, Kaitlyn Bushfield, Orlando Sablon, Nishit Shetty, Linsey C. Marr, Scott E. Hensley, Seema S. Lakdawala.

**Project administration:** Betsy Hunter-Binns, Marlene K. Wolfe, Seema S. Lakdawala.

**Resources:** Betsy Hunter-Binns, Orlando Sablon, Nishit Shetty, Linsey C. Marr, Edith S. Marshall, Kevin Abernathy, Adam S. Lauring.

**Supervision:** Matthew D. Pauly, Scott E. Hensley, Kevin Abernathy, Adam S. Lauring, Marlene K. Wolfe, Jason Lombard, Seema S. Lakdawala.

**Validation:** Kayla Brizuela.

**Visualization:** Abigail P. Paulos, Michelle N. Vu, Chloe Stenkamp-Strahm, Nirmal Kumar, Jenna J. Guthmiller, Seema S. Lakdawala.

**Writing – original draft:** A. J. Campbell, Jason Lombard, Seema S. Lakdawala.

**Writing – review & editing:** Meredith Shephard, Abigail P. Paulos, Matthew D. Pauly, Blaine T. Melody, Marlene K. Wolfe.

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
