## [Editor Report · Decision Letter 0]

30 Oct 2025

Dear Seema,

Thank you for submitting your manuscript entitled "Surveillance on California dairy farms reveals multiple sources of H5N1 transmission" for consideration as a Research Article by PLOS Biology.

Your manuscript has now been evaluated by the PLOS Biology editorial staff and I am writing to let you know that we would like to send your submission out for external peer review. As we were not able to recruit an Academic Editor prior to invite the full submission, there might be some comments from the Academic Editor together with the reviewers.

Once your full submission is complete, your paper will undergo a series of checks in preparation for peer review. After your manuscript has passed the checks it will be sent out for review. To provide the metadata for your submission, please Login to Editorial Manager (https://www.editorialmanager.com/pbiology) within two working days, i.e. by Nov 01 2025 11:59PM.

Kind regards,

Melissa

Melissa Vazquez Hernandez, Ph.D.

Associate Editor

PLOS Biology

---

## [Decision Letter · Decision Letter 1]

14 Jan 2026

Dear Seema,

Thank you for your patience while your manuscript "Surveillance on California dairy farms reveals multiple sources of H5N1 transmission" was peer-reviewed at PLOS Biology. It has now been evaluated by the PLOS Biology editors, an Academic Editor with relevant expertise, and by four independent reviewers.

In light of the reviewers’ reports and our discussion with the Academic Editor, we would like to invite you to revise the manuscript to address the concerns raised. Overall, the assessments agree that the topic is important and the dataset is potentially valuable, but they also identify several issues that must be resolved before we can consider the work further. Most notably, the need to communicate more precisely what the data do and do not show about transmission, and to clearly distinguish evidence for infectious virus in milking-parlor air (more consistent with aerosolization associated with milking/milk) from the RT-qPCR signals reported for exhaled breath, as these imply undamentally different transmission modes, and the quality of data supporting each is very different. Reviewer 1 requests improved transparency and completeness around sequencing and data availability (including functional repository links and accession numbers), and a clearer description of genome/variant calling and coverage. Reviewer 2 is positive with only minor concerns, noting a key unresolved conceptual issue regarding how alternative exposure routes could plausibly lead to udder infection. Reviewer 3 raises concerns about potential pre-PCR contamination and assay comparability (including use of extended cycling and limited negative-control reporting), requests confirmatory H5-specific testing where relevant, and asks for clearer, more fully documented reporting of infectious virus assays (plaque vs TCID50 and supporting raw evidence for low-titer results). Finally, a fourth reviewer raised ethics/reporting concerns and declined to provide a full technical assessment at this stage; while your clarification appears to address the compliance issue, you must correct the Methods, include a complete ethics statement, and carefully check for additional reporting inaccuracies.

IMPORTANT: after consultation with the Academic Editor, we require that you revise the framing in the title/abstract/conclusions to avoid overinterpretation regarding transmission and to align claims with the strength of evidence, and that you address the methodological and reporting issues highlighted above, as these changes are necessary to strengthen confidence in the findings.

Given the extent of revision needed, we cannot make a decision about publication until we have seen the revised manuscript and your response to the reviewers' comments. Your revised manuscript is likely to be sent for further evaluation by all or a subset of the reviewers.

**IMPORTANT - SUBMITTING YOUR REVISION**

*Re-submission Checklist*

*Published Peer Review*

*PLOS Data Policy*

*Blot and Gel Data Policy*

Sincerely,

Melissa

Melissa Vazquez Hernandez, Ph.D.

Associate Editor

PLOS Biology

REVIEWERS' COMMENTS

Reviewer #1 (Ari Machtinger and Dave O'Connor): The authors conducted multiple types of environmental surveillance on dairy farms in California to look for evidence of HPAI H5N1 viral RNA, live virus, and antibodies. They suggest that the evidence of what they detect in the environment reveals multiple potential sources of transmission. This evidence includes detection of H5N1 viral RNA in the air from aggregated exhaled breath samples and from the air in milking parlors, and infectious virus from the air in milking parlors and from water from farm wastewater sources. The authors also demonstrate the presence of variants that are potentially concerning for cow to human zoonosis, that cows can have H5N1 antibodies without the common symptoms of mastitis suggesting subclinical infection, and that the udders of cows do not demonstrate patterns of infection that would be consistent with transmission due to shared milking equipment. All together, they show that transmission may occur by shared milking equipment, by shared/repurposed wastewater, and by aerosol transmission from the breath of cows or due to the milking process. We have three major comments and several minor comments that we recommend the authors consider to improve the manuscript.

Major comments

- The DOI links are not functional as listed for data availability and code availability. We could not tell whether this was due to embargo or a repository issue. In the event that the manuscript is published, raw and interpreted datasets, as well as transformation scripts, need to be made available.

- The SRA accession numbers must be listed for the sequencing datasets.

- The whole genome sequencing for variants is not sufficiently described. What was the method for classifying variants? It is unclear what is meant by "Due to variable sequencing coverage across the genome, mutations were only identified if they were found in each of a sample's sequences from the three primer set amplifications that had sequencing read coverage at that genetic locus." How is lack of variation differentiated from lack of sequence coverage? The authors should consider making the analysis a more significant part of the main text, and potentially include Supplementary Data Table 7 as a main figure with more text to address the above questions.

Minor comments

- The abstract is vague for findings that seem important to clarify. It notes that "Virus was detected in the air in milking parlors and from exhaled breath of cows" but does not clarify that for exhaled breath it was viral RNA, but that for milking parlors it was both viral RNA and infectious virus. It also notes that "sequence analysis revealed viral variants on a farm in these locations" but not that these variants seem relevant for potential human susceptibility. It further notes that "a heterogeneous distribution of infected quarters that maintained a consistent pattern over time" but not that the pattern is inconsistent with transmission due to shared milking equipment, thereby suggesting other modes of transmission may be occurring. These points seem like some of the main takeaways from the manuscript and it would benefit the clarity to directly explain them in the abstract.

- It is not clear what the motivation is for performing exhaled breath sampling from cows. Is it known whether virus-laden aerosols are generated from the cow respiratory tract? And if not, was this the motivation?

- The first sentence of the results section reads "During the winter of 2024". If this date range were clarified to include the months or even days it would help provide more context for the reader.

- Figure 1E shows superior performance of the MD8 Airport over the AirPrep Cub, but there is no elaboration on why this would be the case which is especially important given that the figure also demonstrates that the Cub is operating at a higher flow rate. Why would this result have occurred? Is it because of the mechanism of capture, because of the assay, or some other reason? Should these results even be compared, and were the methods of sampling the same when using both instruments (in the text it described exhaled breath sampling for the MD8 but not the Cub)?

- Air sampling is described in milking parlors and housing pens, but as a reader it is difficult to understand the nature of these environments. To what extent are they indoors/outdoors? How much outside air or sunlight do they receive? Could some of this contribute to dilution or degradation of viruses that might otherwise be expected?

- There is detection of live IAV in wastewater which is not necessarily normal for other wastewater studies. Can the authors comment on why this could be the case? What level of dilution could be happening that explains why we either should or shouldn't expect live viruses?

- For whole genome sequencing, there is missing information which would be useful to interpret the results. What was the depth and coverage of sequencing?

- Supplemental figure 3 shows the "Presence of anti-H5 HA antibodies in expressed milk from cows with subclinical symptoms". This seems like a major finding of the paper, fitting as a main figure rather than a supplementary figure.

- The text mentions "extended" tables multiple times. If this refers to supplementary tables then these should be listed as "supplementary tables".

-- Ari Machtinger and Dave O'Connor

Reviewer #2:

In a comprehensive, data-driven study, the authors concluded that, in addition to virus-positive raw milk, there must be other potential transmission routes for HPAIV H5N1 in US dairy herds. To this end, they carried out detailed analyses of the air and wastewater in affected herds. Individual animals were also examined. It was found that even animals that appeared clinically sound seroconverted, and that animals that were virus-positive in milk did not necessarily develop mastitis. In animals affected by clinical mastitis, not all udder quarters were necessarily affected.

The authors derived meaningful mitigation strategies from these findings: Milkers should wear protective clothing that prevents exposure to aerosols, such as goggles and a mask. Milking equipment should be disinfected after each cow. Milk from infected animals should be disinfected before being passed on to the wastewater system. Subclinically infected animals should be identified using a simple, rapid test that can be carried out on the farm.

To date, experimental studies have shown that direct inoculation of the virus into the udder is necessary for infection. Infection via other routes (intranasal or intramuscular) did not lead to udder infection, and other organ systems are also spared from manifest infection. In this respect, it remains difficult to imagine how udder infections can be established through these alternative infection routes. The authors already envisaged contacts of (injured) teats with virus-positive waste water. Such options should be explored further in their otherwise highly interesting manuscript.

Reviewer #3:

This study from Campbell and colleagues describes surveillance conducted on California dairy farms with H5N1 detections during October, 2024-March, 2025. Both viral RNA and (in a subset of samples) infectious virus were sampled in a range of environmental locations, including the air of housing and milking pens, the exhaled breath of cows, various locations in the wastewater stream of infected farms, as well as longitudinal samples of milk from individual quarters of infected cows. The study detects the presence of viral RNA in the air of milking parlors on farms with confirmed H5N1 outbreaks (determined by the detection of viral RNA in bulk milk tanks), and in multiple reclaimed wastewater sites. The study also shows the presence of anti-H5 antibodies in the milk of dairy cows that were displaying subclinical symptoms at the time of sampling. Finally, the study shows that, in a subset of 9 longitudinally sampled cows, many cows are not positive in all four teats, suggesting (as others have experimentally reported) that infection does not automatically spread from quadrant to quadrant in an infected cow. The question of how transmission of H5N1 influenza A virus is occurring amongst dairy cattle is of high significance for the field and has critical public health implications. The data here suggest that there is abundant viral RNA in many locations on infected dairy farms (particularly in milking parlors, and in wastewater streams). This is not necessarily surprising, as high levels of viral titers have previously been detected in milk of infected animals, and milk from animals known to be positive is generally discarded in the farm wastewater stream rather than being sent to the commercial milk supply. However, the title and conclusions of the manuscript go significantly beyond the data presented in this study, which does not answer the question of how transmission is occurring. Showing that viral RNA is in the air does not prove that transmission occurs through this route. In addition, there are concerns about the lack of negative controls, the fact that methods to prevent pre-PCR contamination are not described, and differences in interpretation regarding whether viral RNA detected in the air is coming from splashes of milk or is being exhaled by cows.

Major Points

* The study (title and conclusions) appears to be drawing a link from viral RNA in the air, concluding this comes from viral RNA exhaled by cows, and then extending that to airborne viral transmission. However, H5N1 in the air in milking parlors as a side effect of the milking process is quite different from H5N1 in the air being breathed out by cows. Tt seems clear from this data that viral RNA can be found readily in the air in milking parlors (where there is abundant splashing of highly infectious milk, a scenario that seems likely to produce aerosols). However, the data for viral RNA in the air outside of this scenario seems considerably weaker. A Ct of 40 for RNA in exhaled breath (Fig 1F) or 4-40 genome copies (Fig 2C) is really borderline positive, and raises concerns about pre-PCR contamination.

* No precautions to avoid pre-PCR contamination are described. This is a significant concern, as pre-PCR contamination (from plasmid or viral RNA present in the lab) is a major concern and the difficulty of avoiding and eradicating such contamination was extensively described during the SARS-CoV-2/COVID-19 response. No-template controls should be included on every plate, but a physically separate location from any amplified product/plasmid/viral RNA extractions is likely necessary to prevent contamination. Many of the conclusions described here are based on very high Ct or very low copies of genome, which is particularly worrying for the possibility of contamination. This is especially true with the modification of increasing the cycling from 40 to 45 cycles, which is an unusual change given the possibility of artifacts when extending over 40 cycles.

* The quantification of H5N1 viral RNA levels is crucial to the conclusions of this manuscript, but it is difficult to understand how samples can be directly compared as two different protocols (RNA extraction, RT-PCR reagents and platform, primers) were used throughout. In addition, some samples were detected via the use of a Swine Influenza Virus RNA Test Kit, which presumably relies on a universal IAV primer (since it appears to detect H5, which is not the predominant subtype in pigs). It is possible that the small number of positive (exhaled) air samples could be a non-H5 subtype (such as H1N1 or H3N2 released from a farm worker infected with a seasonal human subtype). Samples should be run with a second independent, H5-specific primer set, and negative controls should be included in the final figures to rule out environmental contamination (ie, from plasmid or viral RNA present in the laboratory environment).

* As both plaque assays and TCID50s are used (sometimes in the same figure), pictures of the raw assays would help considerably in interpretation. This is especially true for samples in which the plaque titer indicates that a very small number of plaques were present (Fig 2D, Fig S2B). In Fig 3C, it should be clarified whether PFU or TCID50 is being used for each data point. For TCID50, it should be clarified whether other causes of CPE can be ruled out (as bacterial contamination would also cause cell death, and it is likely there is a high level of bacteria in the manure lagoon sample in particular).

* Just because contaminated milking equipment does not appear to be the sole determinant of transmission does not mean that alternative mechanisms of cow-to-cow transmission must exist. Transmission could occur entirely via contaminated equipment, but ALSO require a particular set of intrinsic host/cellular determinants (cellular state, local innate immune state, quadrant specific tissue damage, etc). It seems much more likely that animals with damage to their teats are infected through contact with contaminated milking equipment, vs their teats contacting aerosols or wastewater.

* The point that "the lack of a consistent quarter infected across all cows, as observed for mastitis, suggests that these two features are not linked and likely have distinct modes of transmission" seems to be overstating the case. While 6 of 9 cows (from a single farm) had mastitis in the FR quadrant, and the presence of H5 RNA was more equally distributed among quadrants, it could be possible that H5N1 infection is a pre-requisite for mastitis development. The apparent over representation of mastitis in the FR quadrant could also have a mechanical explanation, as frequent and efficient removal of milk is needed to prevent mastitis symptoms from developing. If the suction or efficiency of the milking equipment that was repeatedly used for the FR quadrant was somewhat less than the other three quadrants, that could explain the preponderance of mastitis there.

* The title and general conclusions greatly overstate what can be concluded by the data. The authors have found some evidence that H5N1 RNA can be found in multiple places within dairy farms with infected herds. This does not prove in any way how transmission is happening, and certainly not that it is happening through multiple different routes.

Minor Points

* As this study relies on procedures involving animals it seems there should be an IACUC review/protocol associated with it.

* Hard to compare Ct to genome copies, a standard curve comparison should be included.

* The sequencing data seems as if it could be very valuable, but is underexplored. It is also unclear which mutations (in supplemental data table 7) are detected in primary samples, vs in cell passaged stocks. The methods section indicates that five of the sequenced air samples include a cell passaged stock, but it is unclear which of the samples in Table S7 this corresponds to. There is also reference to mutations that appear but are not maintained, but it is unclear how this conclusion is supported. Are the same cows sampled over time, so individual mutations can be seen appearing and disappearing? Even in this scenario, with such low genome copies captured, it seems entirely likely that mutations could simply not have been captured upon resampling.

Reviewer #4:

Authors conducted individual animal sampling (milk from all quarters, nasal swabs) without ethics approval. This reviewer believes that IACUC approval should have been obtained prior to animal sampling and would like that concern to be addressed before proceeding with full manuscript review.

---

## [Editor Report · Decision Letter 2]

19 Mar 2026

Dear Seema,

Thank you for your patience while we considered your revised manuscript "Surveillance on California dairy farms reveals multiple possible sources of H5N1 transmission" for publication as a Research Article at PLOS Biology. I have taken over its handling during the absence of my colleague Melissa Vazquez Hernandez from the office, in order to prevent unnecessary loss of time.

This revised version of your manuscript has been evaluated by the PLOS Biology editors and the Academic Editor, and we are likely to accept this manuscript for publication, provided you satisfactorily address the following data and other policy-related requests.

- Title: it would need to indicate that the work relates to influenza virus (non-experts will not immediately understand H5N1), so I propose

Surveillance on California dairy farms reveals multiple possible sources of H5N1 influenza virus transmission

- Data provision: In the FigShare deposition, the data labeled "Figure 6" seems to correspond to "Figure 7", and there is no file for "Figure 6". The data labeled "Sup Fig 3" does not seem to correspond to the relevant figure.

In addition, the FigShare entry is labelled "Raw data for figures and results presented in manuscript, 'Surveillance on California dairy farms reveals multiple sources of H5N1 spread'.", but as it also includes code for SFig 1, please indicate this also in the title (perhaps "Raw data and code....").

- You have selected "Early article posting", but we are considering press releasing the work. If we do go ahead with those plans, we may need to opt your article out of the early posting route to give our press office enough time to arrange this. Please let us know if this would be OK with you.

We expect to receive your revised manuscript within two weeks.

*Published Peer Review History*

*Press*

Sincerely,

Nonia

Nonia Pariente, PhD

Editor in Chief

PLOS Biology

on behalf of

Melissa

Melissa Vazquez Hernandez, Ph.D.

Associate Editor

PLOS Biology

---

## [Editor Report · Decision Letter 3]

1 Apr 2026

Dear Seema,

Thank you for the submission of your revised Research Article "Surveillance on California dairy farms reveals multiple possible sources of H5N1 influenza virus transmission" for publication in PLOS Biology. On behalf of my colleagues and the Academic Editor, Daniel Streicker, I am pleased to say that we can in principle accept your manuscript for publication, provided you address any remaining formatting and reporting issues. These will be detailed in an email you should receive within 2-3 business days from our colleagues in the journal operations team; no action is required from you until then. Please note that we will not be able to formally accept your manuscript and schedule it for publication until you have completed any requested changes.

PRESS

Sincerely,

Melissa

Melissa Vazquez Hernandez, Ph.D., Ph.D.

Associate Editor

PLOS Biology
